# The Small Key to the Treasure Chest: Endogenous Plant Peptides Involved in Symbiotic Interactions

**DOI:** 10.3390/plants14030378

**Published:** 2025-01-26

**Authors:** Anna Mamaeva, Arina Makeeva, Daria Ganaeva

**Affiliations:** Laboratory of System Analysis of Proteins and Peptides, Shemyakin and Ovchinnikov Institute of Bioorganic Chemistry, Russian Academy of Sciences, Moscow 117997, Russia; aryamakeeva@gmail.com (A.M.);

**Keywords:** regulatory peptides, symbiosis, rhizobia, nodules, arbuscular mycorrhizal fungi, plant growth-promoting bacteria, microbiome

## Abstract

Plant growth and development are inextricably connected with rhizosphere organisms. Plants have to balance between strong defenses against pathogens while modulating their immune responses to recruit beneficial organisms such as bacteria and fungi. In recent years, there has been increasing evidence that regulatory peptides are essential in establishing these symbiotic relationships, orchestrating processes that include nutrient acquisition, root architecture modification, and immune modulation. In this review, we provide a comprehensive summary of the peptide families that facilitate beneficial relationships between plants and rhizosphere organisms.

## 1. Introduction

Plant-associated organisms such as bacteria, fungi, microalgae, protists, and viruses have co-evolved with plants for millions of years. Interacting with the host, the members of plant-associated microbial communities provide neutral, beneficial, or detrimental effects on plant growth [1]. Therefore, the interactions between plants and soil microbiota could be considered as a balance between a defense response and the recruitment of beneficial organisms. Mycorrhizae, nitrogen-fixing symbiosis, and utilizing plant growth-promoting bacteria (PGPBs) are examples of beneficial interactions. Plants use beneficial rhizosphere organisms for nitrogen fixation, nutrient acquisition, the synthesis of phytohormones and volatiles, antagonism toward potential pathogens, antifungal activity, and the stimulation of the plant immune system [2].

Symbiosis is any type of close long-term relationships between organisms of different species; however, in the narrow sense, it refers to exclusively mutualistic relationships [1]. In this review, we discuss the relationships between plants and beneficial soil organisms, such as mycorrhizal fungi and bacteria [3,4]. These microorganisms are considered to have evolved from saprophytic or weak pathogenic organisms that were unable to overcome the plant immune system [1]. The best-known example of nitrogen-fixing plant–microbe symbiosis is the interaction between Gram-negative nitrogen-fixing soil bacteria from the genera *Rhizobium* and *Leguminosae*, which results in the formation of bacterial nodules. Typically, flavonoid compounds secreted by the plant trigger, in bacteria, the expression of Nod genes (nodulation), which stimulates bacterial invasion inside the root hair, eventually forming an anaerobic symbiosome inside the nodule [5]. Interestingly, regulatory signals in this plant–diazotroph symbiotic dialogue are often species specific [6].

Legumes form two types of nodules: determinant and indeterminate. Indeterminate nodules are common for IRLC legumes (Inverted Repeat-Lacking Clade, featured by the loss of a 25-kilobase inverted repeat in the chloroplast genome), for example, for *Medicago truncatula*, *Vicia faba*, and *Pisum sativum* [7,8]. Such nodules form a developmental gradient, and bacteria consistently exist at various stages of infection development in different zones of the nodule. As a result, they irreversibly become terminally differentiated bacteroids, which actively fix atmospheric nitrogen while losing the ability for dedifferentiation [9,10]. Determinate nodules are found in plants such as common beans and soybeans [11]. They have a spherical shape, lack zonation, and all stages of bacteroid development occur synchronously within them [11]. 

Another form of nitrogen-fixing symbiosis includes interactions with Gram-positive soil actinomycetes of the genus *Frankia*. Over two hundred plant species from the orders *Fagales*, *Rosales*, and *Cucurbitales* are able to establish symbiotic interactions with these bacteria [12,13]. The majority of plants that form symbiotic relationships with actinomycetes are trees and shrubs. These plants typically develop perennial nodules with numerous lobes, each of which is a modified lateral root that lacks a root cap [13].

The most common and ancient type of symbiotic relationship with fungi is arbuscular mycorrhiza (AM) [14,15,16]. Fungi involved in mycorrhizal symbioses provide plants with mineral nutrition elements (primarily phosphorus) and increase their resistance to abiotic and biotic stress factors: drought, salinity, heavy metal pollution, pathogens, and insects [17]. In these symbiotic relationships, plants supply mycorrhizae with carbohydrates produced during photosynthesis [18]. Plants and fungi form arbuscules, which are highly branched symbiotic structures inside root cells that are used for nutrient transport [14,16]. Most terrestrial plant species (70–90%), including forest trees, wild herbs, and many agricultural crops, form AM with fungi from the *Glomeromycota* phylum [19]. 

In recent years, there has been increasing evidence that the microbiome of plant roots also plays a significant role in different aspects of plant growth and response to stress conditions. The main part of the microbiome, known as ‘core’ organisms, is specific for a particular species, transmitted vertically with seeds, being independent of soil and growing conditions [20]. On the other hand, depending on the conditions, the same microbial species can affect the plant both positively and negatively. The mechanisms of growth-promoting effects may be different, including an increase in the bioavailability of nutrients, influence on the synthesis of phytohormones and the production of their mimetics, an increase in resistance to pathogens, the modulation of resistance to abiotic stresses due to the synthesis of osmoprotectors and antioxidants, the facilitation of colonization of the plant by ‘recognized symbionts’, etc. [21,22]. Moreover, this growth-promoting effect may depend not only on bacteria but on soil algae as well [4]. 

Inoculation with symbiotic organisms changes the expression of a large number of genes in plants. The transcription of these genes is nearly unique for symbiotic cells [23]. Around 10% of transcriptomic changes during rhizobial invasion correspond to plant regulatory peptides [24]. They include members of the NCR (Nodule Cysteine-Rich), GRP (Glycine-Rich Protein) and SNARP/LP (Small Nodulin Acidic RNA-binding Protein/LEED..PEED) families, which are restricted to the IRLC legumes, and CLE (CLavata3/Embryo-surrounding region-related), PSK (Phytosulfokine), and RALF (Rapid Alkalinization Factor) families [23,24,25]. Over 100 Small Secreted Peptide (SSP)-encoding genes change their expression in response to inoculation with rhizobacteria [26]. Among them, there were well-known peptide families involved in nodulation (NCR, PSK, nsLTP—Non-specific Lipid Transfer Protein, SNARP/LP, and GLV—GOLVEN) and peptides that have not been previously shown to be involved in symbiotic regulation (IMA—Iron Man, IDA—Inflorescence Deficient in Abscission, and EPFL—Epidermal Patterning Factor-Like). The transcriptomic response of *Lotus japonicus* to the AM fungus *Rhizophagus irregularis* includes *DEF* (Defensin), *RALF*, *CLE*, and *nsLTP* genes [27]. 

The advantages of peptides as signaling molecules are that they can be very diverse and specifically regulate various processes, the energy costs of their production are relatively low, their expression is easy to regulate, their activity can be modulated via post-translational modifications (PTMs) and proteolysis, and they can be exported from the cell and form a concentration gradient [28]. SSPs are exported from cells in the same manner as proteins [29,30]. However, the localization and trafficking of SSPs have been studied less than for larger proteins and require further study. 

Plant peptides are divided into three classes: released from specific protein precursors; hidden into functional proteins (cryptic peptides); and encoded by small open-reading frames (smORFs) [29,31,32,33,34] (Figure 1). The first class, which includes the majority of known peptide families, is further subdivided into three categories: cysteine-rich peptides (CRPs), PTM peptides, and non-CRP and non-PTM groups. Cryptic peptides are still not shown to be involved in symbiosis establishment. However, taking into account the methodological difficulties of their study and, consequently, the modest number of characterized cryptic peptides in plants, such a picture may not reflect the real state of affairs. In our review, we are focused on the role of regulatory peptides in the establishment of plant symbiotic interactions. 

## 2. Peptide Families Involved in Symbiotic Interactions 

### 2.1. PTM Peptide Families

#### 2.1.1. CLE

CLE peptides take part in plant development processes. They maintain the apex meristem by regulating a delicate balance between cell proliferation and differentiation and participate in the development of vascular tissues [35]. They are also represented in the genomes of *Physcomitrium* and *Marchantia*, so CLE peptides probably have evolved from the last common ancestor of land plants [36]. Active 12–13amino acid (a.a.) CLEs are cleaved from 80–100 a.a. pre-propeptides, which consist of an N-terminal signal peptide, a variable domain, and a highly conserved C-terminal CLE domain (Figure 2A) [37]. CLE signal perception is a complex process with multiple membrane receptors: LRR-RLKs (CLAVATA1—CLV1; Hypernodulation Aberrant Root formation 1—HAR1; Super Numeric Nodules—SUNN), and coreceptors such as receptor-like protein (CLV2) and pseudokinase CORYNE [38,39,40]. 

Certain *CLE* genes are expressed in legumes in response to rhizobacterial infection and the presence of nitrogen in the soil, thereby triggering AON (Autoregulation Of Nodulation), a systemic regulatory mechanism that inhibits the formation of an excessive number of nodules [35]. Nodulation-suppressing CLE peptides may act redundantly and can be subdivided to only rhizobial-induced and both rhizobia and nitrate-induced groups [43,44,45]. *CLE* genes are expressed at early stages of nodule development in nodule primordia and developing nodules, both in IRLC and non-IRLC legumes [46,47]. Also, CLEs are expressed in apical zones of mature indeterminate nodules of IRLC legumes [46]. The effect of CLE peptides on the root system is systemic: *MtCLE12*, *MtCLE13*, and *MtCLE35* overexpression (OE) in some roots inhibits nodule formation in the whole root system of the plant [46,48]. Curiously, a deeper analysis of MtCLE35 effects at the transcriptional level reveals both systemic (induction of Too Much Love1/2—TML1/2) and local changes in the downstream CLE signaling pathway (SUNN, TRX—thioredoxin, and D53) [49]. It has been previously shown that CLE increases the expression of *TML1/2* and induces AON by decreasing the expression of its repressor, mobile miRNA *miR2111* [50,51]. The transcriptomic analysis of *MtCLE35* OE plants reveals that *Mt*CLE35 arrests the rhizobacterial-induced expression of regulators of nodule development [52]. 

CLEs can undergo two types of modifications: proline hydroxylation and arabinosylation of hydroxyproline, which is carried out by glycosyltransferases from the HPAT family (hydroxyproline O-arabinosyltransferase) [40]. CLE’s modifications often, but not always, are essential for physiological activity [51,53,54]. *Mt*CLE12-13 triggers AON in the triarabinosylated form but not in the monoarabinosylated or unmodified form [55]. HPAT PLENTY presumably performs the arabinosylation of hydroxyproline in CLE-RS (CLE-Related Root Signal) molecules of *L. japonicus*, which is necessary for nodulation inhibition activity [56]. Another enzyme from the HPAT family, *Mt*RDN1 (Root Determined Nodulation 1), is localized in the Golgi complex and is required for AON induction by *Mt*CLE12 and *Mt*CLE35 but not *Mt*CLE13 [44,57]. 

CLEs are also involved in the regulation of interactions with mycorrhizal fungi. In particular, *MtCLE53* expression is reduced under phosphorus deficiency and is induced both by phosphate and AMF (arbuscular mycorrhizal fungi) colonization [58]. AMF colonization induces the expression of three more CLE genes: *MtCLE16*, *MtCLE43*, and *MtCLE45*. The *Mt*CLE53 peptide SUNN dependently inhibits the synthesis of strigolactones [59] which are synthesized in plants under phosphate deficiency and are involved in the regulation of plant–mycorrhizal fungus interactions [60,61]. Similarly, CLE peptides play an important role in non-legume plants. For instance, in tomatoes, *Sl*CLE11 inhibits mycorrhiza development [62]. It is assumed that *Sl*CLE11 works in the early stages of mycorrhiza development, and arabinosylation is a prerequisite for activity. Moreover, FAB (LRR-RLK) and FIN (O-arabinosyltransferase) proteins are negative regulators of mycorrhiza formation and participants of AOM (Autoregulation Of Mycorrhizae) [63]. Interestingly, these proteins are involved in the inhibition of symbiosis development with fungi by nitrate but not by phosphate. FAB and FIN inhibit symbiosis at the early stages, affecting the number, rather than the structure, of arbuscules. At the same time, CLE peptides were detected in the genomes of mycorrhizae-forming basidiomycetes: *Rhizophagus irregularis* and *Gigaspora rosea*. It was shown that these peptides influence the growth of plant roots and stimulate the development of mycorrhiza [64]. 

#### 2.1.2. CEP

C-terminally encoded peptides (CEPs) are a multigene family present in seed plants. In general, CEP peptides orchestrate the root system architecture and nutrient uptake [65]. The CEP precursor consists of an N-terminal signal peptide, variable domain, and one or several CEP domains, whereas mature CEPs are a 15 a.a. PTM peptide (Figure 2B) [66]. 

CEPs are synthesized in the roots during nitrogen deficiency and stimulate nodule development [67]. A total of 6 out of 21 *PsCEPs* exhibit high transcription levels in roots and nodules [68]. *MtCEP1* expression is induced by low nitrogen concentrations and high concentrations of CO_2_ [69]. Both OE and exogenous *Mt*CEP1 treatment promote nodulation. Such plants produce larger nodules, which develop faster and fix more nitrogen [69]. *Mt*CEP1 induces significant transcriptome changes, including the transcription of leghemoglobin genes and the repression of defense genes [69]. Notably, *GmCEP6*, which is also induced by inoculation with rhizobacteria and inhibited by nitrate, stimulates nodulation and NF (Nodular Factor) pathway marker gene expression, including *GmENOD40* (Early nodulin 40), *GmNINa* (Nodule Inception), and *GmNSP1* (Nodulation Signaling Pathway 1) [70]. The regulation of *Mt*CEP1-mediated nodule formation occurs via LRR-RLK *Mt*CRA2 (Compact Root Architecture2), a homologue of the CEPR receptor in *Arabidopsis* [71]. *Mt*CRA2 phosphorylates EIN2 (Ethylene Insensitive 2), thereby alleviating the inhibition of ethylene on nodule formation [71].

CEPs, as important mineral nutrition regulators, also modulate plant–fungus symbiosis. Tomato inoculation with AM fungus *R. irregularis* suppresses *SlCEP2* expression [72]. *SlCEP2* downregulates proteins involved in synthesis, transport, and signaling of auxins, which leads to a decrease in the number and density of lateral roots. This process is mediated by *Sl*CEPR1, an LRR-RLK that is an orthologue of the CEP receptors *At*CEPR1 and *Mt*CRA2. The reduction of *SlCEP2* expression upon AM fungus inoculation results in increased lateral root formation [72]. Otherwise, *MtCEP1* is induced by low phosphorus conditions and promotes AM symbiosis [73]. The mutation of the *Mt*CRA2 receptor decreases strigolactone biosynthesis and the expression of phosphorus-responsive and AM-related genes, indicating that the CEP1/CRA2 pathway regulates AM symbiosis establishment.

To conclude, CLE and CEP groups both regulate the extent of nodule formation through antagonistic mechanisms. They depend on the availability of macronutrients and exhibit a fine-tuned regulatory system of nodulation, taking into account plant needs and resources [67,74]. They both regulate miR2111, a mobile miRNA that represses the AON component TML1/2: CEPs promote *miR2111* expression and nodulation, while CLE peptides show opposite effects [51,75]. MiR2111 is not the only link between CEP and CLE. Transcription factor *Mt*NIN, which regulates interactions with rhizobacteria, has been shown to bind to the promoters and stimulate the expressions of *MtCEP7* and *MtCLE13* [76]. The coordinated regulation of these peptides’ expressions allows for the control of nodulation not only in response to the availability of mineral nitrogen but also with consideration for the nitrogen-fixing efficiency of the symbiotic strain [67].

#### 2.1.3. RGF

GLVs or RGFs (Root Meristem Growth Factors) regulate multiple processes, including root meristem maintenance, the development of root hairs, lateral roots, and gravitropism (Figure 2C) [77]. Members of this family are found in all land plants.

GLV peptides are involved in the regulation of beneficial interactions with rhizobia. At least 5 of the 15 RGFs are responsible for the concerted control of nodulation during symbiosis in *Medicago* [78,79]. Two *MtRGF* orthologues in non-IRLC *Soybean* demonstrated nodule-specific expression too [79]. 

*MtRGF5/GLV10* is expressed specifically in dividing cells underlying infection sites. This peptide takes part in nodule positioning along the primary root axis: exogenous GLV10 treatment induces a shift in the position of the first nodule, moving it from the root’s midpoint to approximately 60% of the root length [79]. 

*MtRGF3/GLV7* is expressed in nodules in response to the Nod-factor secreted by *Sinorhizobium meliloti* and negatively regulates nodule development [78]. Moreover, it inhibits the expression of key nodulation regulators (*NFP*—Nod Factor Perception, *LYK3*—Lysin motif receptor-like Kinase3, *ERN1*–Required for Nodulation1, *NSP2*, and *NIN*) [80]. It also activates an immune response, leading to the inhibition of rhizobacterial infection [73]. Meanwhile, leaf-expressed *GLVs* take part in immune reactions via the regulation of PRR levels: the overexpression of *RGF6(GLV1)* and *RGF9(GLV2)* increases flg-22 responses [81]. 

#### 2.1.4. PSK

PSKs are Tyr-sulfated PTM pentapeptides (Figure 2D) that regulate cell proliferation and differentiation, root growth, abiotic stress response, and innate immunity [82,83]. PSK peptides cleaved from 80 to 120 a.a. precursors having an N-terminal signal peptide. Mature peptides are recognized by the plasma-membrane LRR-RLK PSKR (PSK Receptor) [82].

In *Lotus japonicus*, five *PSK* genes were identified, three of which (*LjPSK1*, *LjPSK4*, and *LjPSK5*) are expressed in nodules [84]. *LjPSK4* expression was also induced in spontaneous nodules, whereas the induction of *LjPSK1* required rhizobacterial infection. The promoter activity of *LjPSK1* was observed throughout the nodule, with increased activity in infected cells. The overexpression of *LjPSK1*, as well as the PSK1 receptor from Arabidopsis (*AtPSKR1*) led to an increase in the number of nodules [84]. 

PSK-δ is a pentapeptide identified specifically in legumes such as *M. truncatula*, *L. japonicus*, and *G. max* [85,86]. In *M. truncatula*, the PSK-δ-coding gene is expressed in cells that initiate nodule formation, and its expression is maintained in nodules until maturity, whereas mutant lines studies (*PSK*-δ OE and *PSK*-δ KO) along with RNAi and PSK-δ treatment demonstrated that PSK-δ stimulates nodule formation [85,87]. 

PSK-ε is another legume-specific disulfated pentapeptide that differs from PSK-δ by a single a.a. PSK-ε has also been identified in *M. truncatula*, *L. japonicus*, and *G. max* genomes. *MtPSK-ε* is expressed in the root apex and lateral root primordia. Its expression is also induced by rhizobacterial inoculation and is maintained in the primordia and apical zone of developing nodules. Treatment with exogenous *Mt*PSK-ε stimulates root growth, lateral root formation, and nodule development [86].

The *At*PSKR positively and specifically regulates the colonization of the rhizosphere by *P. fluorescens* [88]. Knockout *AtPSKR1* plants during *P. fluorescens* inoculation demonstrate inhibited root growth and increased transcription of SA-mediated defense genes. Inoculation with *P. fluorescens* enhances the expression of *AtPSKR1* in the roots. Presumably, an elicitor produced by *P. fluorescens* that modulates plant defense responses is not a PSK mimetic [88].

Overall, among PSKs, there are both common peptides shared by different plant taxa and legume-specific peptides that regulate symbiotic interactions [84,85,86,87]. Their involvement extends beyond nodulation, influencing root growth and rhizosphere colonization, highlighting the potential of PSK pathways as targets for enhancing plant–microbe interactions in agricultural contexts [89]. 

### 2.2. Cys-Rich Peptide Families

#### 2.2.1. NCR

NCR is a large CRP family, specific to nodules and exhibiting high sequence and isoelectric point variation within the family [90]. Members of this family regulate the formation of indeterminate nodules. NCR peptides have not been discovered in legumes with a determinate type of nodulation [7]. This peptide family is phylogenetically related to defensins [91], which are widespread in both plants and animals [92]. Individual defensins have been shown to participate in the interaction of plants with symbiotic fungi [93,94]; however, their main function is defense response against pathogens [95]. Mature NCRs consist of 35–55 a.a. and contain 4 or 6 conserved cysteine residues, whereas defensins contain 8 or 10 cysteines forming disulfide bounds (Figure 2E–G) [90,96]. It is shown that all four conserved Cys residues in NCR169 and NCR343 peptides are required for biological activity [97,98].

Like defensins, NCRs exhibit antimicrobial and antifungal activities [90,96]. For some NCRs, for example, *Mt*NCR169, *Mt*NCR211, and *Mt*NCR343, antimicrobial activity against free-living rhizobacteria is observed [98,99]. Notably, chimeric NCR247 peptide activity might be comparable to third-generation antibiotics [100]. The NCR169C17-38 fragment demonstrates significant inhibiting influence on ESKAPE pathogens (*Enterococcus faecalis*, *Staphylococcus aureus*, *Klebsiella pneumonia*, *Acinetobacter baumannii*, *Pseudomonas aeruginosa*, and *Escherichia coli*) [101]. Whether this antimicrobial activity is crucial for restriction of rhizobacteria in nodules still requires further evaluation.

Genomes of different IRLC plants contain from several to several hundred *NCR* genes [5,6]. The number of transcribed *NCRs* in different legumes correlates with the degree of bacteroid elongation [7,102]. Different NCR peptides vary in their expression patterns, depending on stages of development and localization in the nodule zones [7]. NCRs regulate the process of terminal differentiation of bacteroids. For instance, *MtNCR086* and *MtNCR314* knockout (KO) mutants develop small, round nodules that are populated by rhizobacteria. On the other hand, they do not increase in size, do not undergo endoreduplication, and fix nitrogen [103]. The peptides MtNCR169, MtNCR211, MtNCR343, and MtNCR-new35 are also essential for bacteroid differentiation and/or maintenance [97,98,99,104]. The mutation of each of these genes leads to inefficient nitrogen fixation, incomplete bacteroid differentiation, the expression of senescence-associated genes, and cell death in the nitrogen fixation zone of the nodule. Also, some NCRs control discrimination against rhizobacterial strains by provoking bacterial cell death and early nodule senescence in a strain-dependent manner [105,106].

Exogenous treatment of recombinant NCR peptides in *Astragalus sinicus* revealed that different *As*NCRs can stimulate, inhibit, or even have no effect on the growth of rhizobacteria in vitro [107]. However, not all NCR peptides are essential for establishing symbiotic relationships. Thus, the CRISPR/Cas9 knockout of the *MtNCR068*, *MtNCR089*, *MtNCR128*, and *MtNCR161* genes has no effect on the interaction between *M. truncatula* Jemalong and *Sinorhizobium medicae* WSM419 and the formation of nodules [108]. This may also suggest functional redundancy among NCR peptides or the involvement of these NCR peptides only in several plant–rhizobium pairings to govern the specificity of symbiotic interaction.

Notably, the activity and stability of NCRs can be influenced by the redox status and position of disulfide bridges in the molecule [109]. Thioredoxin s1 (trx s1), which is specific to nodules, is able to reduce S-S bonds in NCR247 and NCR335 molecules. Moreover, these peptides exhibit stronger antimicrobial activity in the reduced state, so the presence of trx s1 is necessary for the terminal differentiation of bacteroids [110]. This way, a complex system is formed, which allows the plant to fine-tune its interaction with rhizobacteria to efficiently exploit them.

It is generally believed that NCR concentrations that would be sufficient to cause serious damage to nodule bacteria are not reached in nodules [7,111]. Moreover, nodule bacteria have a number of adaptation strategies to cope with the toxic effects of these peptides. BacA, a member of the ABC transporter group, defends rhizobia, probably by exporting NCR peptides from bacteroids [10]. Both HrrP (Host range restriction peptidase) and SapA (Symbiotic associated peptidase) participate in the cleavage and detoxification of NCR peptides; however, they differ in substrate specificity [112]. HrrP is present in about 10% of *S. meliloti* strains [113]. *HrrP* expression could lead to less pronounced host plant control and the suppression of nitrogen fixation [114]. Based on *Medicago* insights, it is proposed that the combination of peptides and peptidases in a plant–bacterium system ensures strict control of the dosage and composition of NCRs and determines the specificity of symbiotic relations [115]. In this case, NCR peptides modulate the physiological state of bacteria by inducing terminal differentiation. For example, NCR247 at sublethal concentrations alters the expression of 15% of *S. meliloti* genes, including key cell cycle regulators, which leads to a blockage of cell division and endoreduplication [116]. Recently, NCR247 was found to induce additional iron uptake required for symbiotic nitrogen fixation [117]. NCR247 is the first plant peptide with an established role in the regulation of metal metabolism in symbiotic organisms. These results also demonstrate a new aspect of NCR-peptide functioning.

For a long time, NCR peptides were thought to be present only in the IRLC clade, but it has been found that NCRs are also present in the genus *Aeschynomene* [118]. Curiously, in these plants, nodulation occurs in a Nod-factor independent manner and without the formation of infection threads. Moreover, they lack several genes that are involved in the regulation of nodulation in other legumes (*NFH1* and *EPR3*). The NCR of *A. evenia* can be divided into two groups: NCR with Cys-rich motif 1, similar to *M. truncatula* NCRs (26 genes), and defensin-like motif 2 (32 genes) [118]. At the same time, in other species of the genus—*A. duranensis* and *A. ipaiensis*—only genes belonging to the second group were found, and for most of them, their expression is induced in nodules. Thus, the nature of the appearance and evolution of NCRs still remains unclear.

NCR peptides are specific to a narrow group of leguminous plants. A great variety of NCR peptides has been discovered, but there is still a lack of a complete understanding of their functions. NCRs are known to be essential for the correct establishment of symbiotic relationships. They may have strong antimicrobial activity and trigger the terminal differentiation of bacteroidetes. It has been suggested that NCR peptides are involved in post-infection symbiotic specificity between legumes and nodules [10]. Data from a large-scale transcriptome study indicate that enhanced control of the bacterial symbionts gained by the expansion of putative antimicrobial peptide gene families allows NCRs to be conserved during the evolutionary process [119]. 

#### 2.2.2. DEF

DEFs are immune peptides, which take part in the establishment of symbiotic relationships with fungi [93]. DEF genes are upregulated in *M. truncatula* upon mycorrhizal fungi inoculation, and their expression correlates with AM marker genes (*MtPt4* and *MtMyb1*) [93,94]. Neither the knockdown nor OE of *MtDefMd1* and *MtDefMd2* influenced arbuscula size; additionally, the knockdown of both genes did not suppress AM marker gene expression. Likely, DEFs play a role in late stages of arbuscula development, during turnover into a post-symbiotic cell [94]. DEFs play an important role both in pathogenic fungi resistance and the maintenance of symbiotic arbuscula development, and these functions do not overlap. For example, *MtDef4.2* OE in wheat led to plant resistance against *Puccinia triticina* but did not affect symbiotic interactions with *Rhizophagus irregularis* [120].

#### 2.2.3. RALF

RALFs are Cys-rich SSPs that affect roots and pollen tube growth, nodulation, and stress resistance [121,122,123]. Moreover, RALFs modulate the immune response, and different members of the RALF family can either stimulate or suppress it [124,125]. The mature RALF peptide typically contains an RR motif, highly conserved YISY motif, and four conserved cysteine-forming disulfide bonds (Figure 2H) [126,127]. The receptors of RALF peptides belong to the *Catharanthus roseus* receptor-like kinase 1-like (*Cr*RLK1L) family and include such proteins as FERONIA (FER), THESEUS 1 (THE1), etc. [126,128].

In *Phaseolus vulgaris*, the expressions of *RALF* and *FER1* are induced by rhizobacterial colonization and persist in nodule primordia throughout nodule development [129]. This effect is nitrogen dependent: in the absence of nitrogen, *Pv*RALFs promote nodulation; however, in the case of nitrogen presence in the growth medium, peptides suppress this process by inducing AON [129]. In addition, *RALFL1* expression is upregulated by Nod factors in *M. truncatula*, while the overexpression of this gene causes the development of small defective nodules [130]. The analysis of *RALF* gene expression in *Glycine max* infected with *Bradyrhizobium japonicum* showed that the transcription of 6 genes was downregulated, while 10 were upregulated [131]. This discrepancy in expression patterns suggests the potential dual role of RALF peptides, with downregulated genes contributing to the immune response and upregulated genes supporting symbiotic relationships. Although intriguing, these roles remain speculative and require experimental confirmation.

The RALF-FERONIA system interacts with beneficial rhizosphere microbiomes. In *FER1 knockout* mutants, the root microbiome is enriched with *Pseudomonas fluorescens*, which is a PGPB [132]. A similar effect was observed with the exogenous treatment and OE of *RALF23*. Further studies showed that RALF23 accumulates and interacts with FER1 under phosphate deficiency, resulting in the disruption of the FLS2-BAK1 complex and the suppression of plant immunity [124,133]. This leads to the enrichment of the root microbiome with *Bacillus* and *Pseudomonas*, which helps plants adapt to phosphate deficiency [133].

RALF peptides are widely represented among different plant taxa from *Physcomitrella* and *Selaginella* to angiosperms [127]. Curiously, some rhizosphere organisms acquired *RALF* genes via horizontal gene transfer from their hosts [134]. For instance, a peptide from the root endophyte *Colletotrichum tofieldiae*, *Ct*RALF, interacts with plant receptor FER1, which further enhances their symbiotic interactions [135]. Conversely, pathogenic nematodes and fungi use RALF mimetics to inhibit the immune response [136,137]. 

Taken together, RALF peptides orchestrate the relationship between plants and rhizosphere organisms, including pathogens and beneficial bacteria. To fully understand differences in RALF involvement in plant responses to symbiotic microorganisms and pathogens, more detailed research is needed [126]. 

#### 2.2.4. nsLTP

nsLTPs represent a large group of CRPs widespread among most plant taxa [138]. nsLTPs have an N-terminal signal peptide and eight cysteine motifs (8CMs), which form four disulfide bridges in the mature peptide. nsLTPs exhibit lipid binding and transferase activity. These peptides are involved in various stress responses, such as drought and heat tolerance, reproduction, and cuticle formation [139]. They also take part in the immune response: *LTP* transcripts accumulate in response to pathogenic infections [140,141].

nsLTPs also play a role in regulating the early stages of symbiosis establishment in both *Rhizobacteria* and *Actinomycetes* [142]. In species capable of legume–rhizobium symbiosis, a significant proportion of *nsLTPs* in their genomes are induced during nodulation [142]. Notably, *Ag*LTP24 from *Alnus glutinosa*, at subinhibitory concentrations, induces transcriptional changes in 142 genes in *Frankia*, including chaperones, ABC transporters, and proteins involved in cell wall/membrane/envelope biosynthesis [142]. 

In legumes, it was shown that members of the nsLTP1 and nsLTPd classes are widely expressed in roots and nodules, confirming that this group is involved in nodulation [138]. *As*E246 (member of nsLTP in *Astragalus sinicus*, expressed exclusively in nodules) binds to lipids in the symbiosome membrane and is localized in infection threads [143,144]. Additionally, the OE of *AsE246* led to an increased number of nodules, while knockout mutant nodule development slowed down: plants had fewer mature infected cells compared to the control [144]. It can be suggested that members of the nsLTP family remodulate membranes in nodules and are involved in interactions between the plant and symbionts. 

### 2.3. Non-Cys Rich, Non-PTM Peptide Families

#### 2.3.1. GRP

GRP is a large peptide superfamily with high glycine content. GRPs are involved in the cell wall structure, cell growth, stress responses, flower development, and pollen hydration [145,146]. Depending on their general structure, GRPs are divided into five classes [147]. Class IV GRPs have an RNA-binding domain and exhibit antimicrobial activity [148,149]. 

In *M. truncatula* and *M. sativa*, GRPs with nodule-specific expression patterns have been identified. These genes were shown to not be expressed in aboveground parts of plants, and neither stresses nor hormones induced their expression, unlike other *GRPs* [150]. The expression of *MsnodGRP1-2* was detected in the interzone, and *MsnodGRP3* in the nitrogen-fixing zone III. These GRPs presumably regulate nodulation or/and bacteroid differentiation, which is most likely not functionally redundant [150]. Similar to the NCR, nodule-specific GRP peptides have been discovered in legumes amongst IRLC members only so far [151]. 

#### 2.3.2. SNARP

SNARPs were discovered in *M. truncatula* as a legume-specific peptide family based on a comparison of sequences in legume and non-legume plants [152]. *Mt*SNARP2 is an early induced nodulin that interacts with *ENOD40* RNA. *Mt*SNARP2 silencing led to a significantly high number of aberrant nodules that had a hypertrophied outer cortex and contained a brown pigment instead of pink. These nodules are characterized by an altered anatomy, a lack of clear zonation, and the fact that rhizobacteria do not remain metabolically active after cell infection [153].

Trujillo et al. (2014) conducted bioinformatics research of genes encoding SNARPs in legumes and other plants (*A. thaliana*, *Oryza sativa*, and *Populus trichocarpa*); however, these genes were discovered only in *Medicago* (*M. truncatula*—13; *M. truncatula* ssp. tricycla—11; *M. sativa*—10). The author proposed renaming the SNARP family to LEED..PEED, as it indicates the conservative motifs of all representatives so far, as RNA-binding activity was detected only in *Mt*SNARP2/LP11 [154]. 

All 13 members of the *MtSNARP/LP* family are exclusively transcribed in nodules; however, they exhibit different expression patterns. More specifically, *SNARP/LP* expression was detected at 4 up to 28 dpi in different zones in nodules starting in the infection zone, ending in the nitrogen-fixing zone [153,154]. Additionally, Trujillo et al. (2014) found *MtSNARP2/LP11* transcripts in the meristem zone. SNARP/LP is a unique family for *Medicago*.

### 2.4. SmORF-Derived Peptide Families

#### 2.4.1. DVL

DVL/ROT (DEVIL-like/ROTUNDIFOLIA) is a peptide family encoded by smORFs (Figure 2I). DVLs are involved in the regulation of leaf and flower development, the shade response, and root growth under abiotic stress [155,156,157,158]. Along with *MtRALF1*, *MtDVL1* is a Nod-factor-upregulated gene. In addition, plants overexpressing *MtDVL1* and inoculated with *S. meliloti* demonstrate a reduction in the number of nodules, although this does not affect nodule development [130]. The smORF-coding peptides are yet to be examined, and only ENOD40 is known to be involved in rhizospheric interactions, apart from the DVL family. 

#### 2.4.2. ENOD40

ENOD40 plays an important role in bacteroid development during rhizobia inoculation in legumes [159,160]. However, *ENOD40* genes are also found in non-legume plants [161]. These genes contain two conserved regions. Region 1 codes short peptides that are 11–13 a.a. in length [161]. It is involved in the regulation of sucrose utilization in nodules and specifically binds to NODULIN100, a subunit of sucrose synthase [162]. Furthermore, ENOD40 transcripts exhibit RNA-mediated regulatory functions [163,164]. 

*Medicago truncatula* codes two *ENOD40* genes [160]. Both *MtENOD40-1* and *MtENOD40-2* are essential for nodule initiation; moreover, the effect of *ENOD40* genes is dose dependent [160]. In soybean, the *ENOD40* gene functions as part of the regulatory module *miR169c-NFYA-C-ENOD40* (NFYA, Nuclear Factor-Y Subunit A) and, depending on nitrogen availability, regulates interactions with rhizobia [165,166]. The overexpression of *MtENOD40* has also been shown to enhance mycorrhizal colonization [167]. Despite its established contribution to symbiosis, the precise mechanisms by which ENOD40 transcripts influence plant–rhizosphere interactions and downstream signaling pathways remain critically understudied. 

## 3. Peptides Involved in Both Symbiosis and Immunity

The regulation of immune response amplitude is crucial for establishing effective symbiotic relationships [168,169]. Rhizobacteria induce a transient immune response in the host plant: initially, upon infection with rhizobia, defense gene expression increases, but it subsequently declines [24,25,168,170]. The immune response triggered by PGPB includes PR gene expression and ROS production; however, the intensity of this response is lower than elicited by pathogens [85]. In addition, the activation of the immune system and ROS production upon the interaction between *A. thaliana* and the PGPB *Bacillus velezensis* FZB42 was shown to be essential for bacterial colonization and auxin synthesis [171]. In turn, the inoculation of *Aeschynomene evenia* by *Bradyrhizobium* does not trigger an immune response, suggesting the different strategies of interactions between symbiotic organisms and plant immune systems [172]. However, our understanding remains limited regarding how plants differentiate between pathogens and PGPB [85].

It has been shown that symbiotic interactions are highly specific, triggering individual metabolic changes at the subspecies level (cultivar strain) [6]. In addition, certain strains have the potential to impede the growth of specific plants, indicating that potentially beneficial interactions may be detrimental. It makes distinguishing between pathogens and symbionts more complicated.

Regulatory peptides play a role in suppressing defense responses during the establishment of symbiotic relationships (Table 1). For instance, members of the RALF peptide family, which can either enhance or suppress immune responses, are involved in the control of symbiotic interactions [129,130]. Besides RALFs, PSKs also participate in this process: recent studies showed that *PSKR* OE demonstrated decreased resistance to pathogens, whereas knockout lines were less susceptible to infections [173]. Beneficial bacterium *P. fluorescens* stimulates the expression of *AtPSKR1*, which inhibits defense genes and facilitates bacterial colonization [88]. Moreover, pathogenic fungi can also manipulate the plant defense system by enhancing the PSK signaling pathway [174]. Additionally, nsLTPs, which regulate plant interactions with diverse nitrogen-fixing organisms, can also promote an immune response and exhibit antimicrobial activity [142,175,176].

Plants secrete metabolites of various types, which exhibit selective antimicrobial activity, thereby modulating the composition of the microbiota [179], for example, NCR peptides [90,96,180]. In addition, certain members of defensins—a large group of protective and antimicrobial peptides—have been found to participate in the establishment of symbiosis [94].

Thus, certain peptide families exhibit functionality in both symbiosis and immunity, acting as critical regulators that fine-tune the balance between growth and defense in plants (Table 1). The peptide families, such as RALF, PSK, and nsLTPs, participate in both the suppression and enhancement of immune responses while promoting symbiotic relationships. 

## 4. Rhizosphere Organisms Mimic Plant Peptides

Plant rhizosphere organisms have evolved mechanisms to manipulate host plants for promoting pathogenic and symbiotic interactions. Plant-interacting organisms manipulate phytohormone levels and signaling pathways to overcome plant immunity and eventually get benefits from host plants. One of the mechanisms these rhizosphere organisms can employ to dampen immune responses and facilitate these interactions is to produce peptides that mimic the host plant peptide hormones involved in immune defenses or root development [181].

Molecular mimicry is a widespread phenomenon where organisms have evolved proteins/peptides that imitate host cell proteins [181,182]. To date, various phytopathogenic nematodes have been found to produce analogues of CLE, CEP, RALF, and IDA peptides, which are essential for infection development [35,137,177,183,184]. Interestingly, while IDA is not currently known to participate in symbiotic interactions, it is possible that future studies may reveal its involvement. Among parasitic fungi, peptide mimetics of the IDA, RALF, PSK, and PEP families are present [136,181,185]. RALF and PSK peptides were identified in more than 20 *Basidiomycota* and *Ascomycota* species. As for pathogenic bacteria, mimetics of plant RALF, CEP, and PSY peptides were found [186]. 

Very few examples of mimetic regulatory peptides have been identified in symbiotic and beneficial microbes, which is intriguing because beneficial organisms, the same as pathogens, need to weaken the plant defense system to engage into interaction [1]. Among *Rhizobacteria*, such peptides have not been discovered so far. At the same time, in other non-pathogenic bacterial species of the *Actinobacteria*, *Proteobacteria*, and *Gemmatimonadetes* phyla, genes encoding possible homologs of the CLE, CEP, PSK, and PEP peptides have been identified [181]. The *Actinobacteria* sp. encodes two potential homologs of plant CLE peptides. In addition, PSK and PEP homologs have recently been identified in two plant-associated, non-pathogenic bacterial species. The PSK homolog containing the C-terminal PSK domain and N-terminal signal domain was identified in *Proteobacteria* sp. isolated from the phyllosphere metagenome [178]. However, the functions of these PEP and PSK are still unknown. Additionally, in the genomes of arbuscular mycorrhizal fungi (four *Rhizophagus* species and one *Gigaspora* species), genes encoding CLE-like peptides have been discovered. Exogenous CLE-like peptide treatment affects the root architecture system and stimulates plant colonization by the corresponding fungi [64]. In addition, symbiotic fungi may produce their own peptides to promote colonization [187,188]. 

In summary, plant peptide mimetics from rhizosphere microbiota are able to bind to receptors of plant peptide phytohormones and enhance the efficiency of the infection process. However, the diversity and mechanisms of this mimicry remain elusive and should be studied further. Moreover, the nature of this mimicry is still under discussion [181]. Whether it is the result of coevolution or horizontal gene transfer remains unclear.

## 5. Common Peptide Families for Different Symbiotic Interactions

There is a growing body of evidence that the rhizosphere microbiome co-evolves alongside its plant hosts [126], with many aspects of plant–microbe interaction regulation being highly conserved [20]. While several hypotheses suggest that regulatory peptides are involved in these processes, our understanding of this mechanism remains extremely sparse [88,129]. Consequently, elucidating the role of peptides in modulating interactions, not only with rhizobial bacteria and mycorrhizal fungi but also with the broader microbiome, represents a timely and ambitious challenge.

Are there common regulators involved in the interactions between plants and members of such diverse phylogenetic groups? It is noteworthy that certain groups of phytohormones often have similar effects on symbiosis with both bacteria and fungi. For instance, auxins and strigolactones act as positive regulators of symbiotic relationships, whereas ethylene plays a negative role [189]. Among peptides controlling symbiotic interactions with both fungi and rhizobacteria, shared participants have also been identified: CLE and CEP (Table 1, Figure 3).

The most extensively studied peptide family in relation to symbiosis is the CLE family. CLE peptides have been shown to play an important role both in AON and AOM [35,58,74,190]. Thus, CLE peptides perform similar functions in regulating the interactions between plants and both rhizobacteria and fungi. Interestingly, it was shown that CLE1 is involved in haustorium formation of parasitic *Phtheirospermum japonicum* [191]. This peptide belongs to group 3A in the CLE family, being essential for AON regulation in legumes [191]. 

NCR peptides, previously considered to be specific to legumes in the IRLC clade, were later identified in a distinctive group (*Aeschynomene* spp.), which lacks several other common regulatory systems, including the Nod-factor recognition system [118]. NCR peptides are related to defensins, which also participate in the regulation of mycorrhizal development [94]. 

PSK peptides promote nodule formation, while the *At*PSKR1 receptor enhances interactions with *P. fluorescens* [84,85,86,87,88]. Besides PSKs, RALF peptides regulate interactions with rhizobacteria and modulate the structure of the rhizosphere microbiome [129,130,132,133]. 

*CEP* expression is upregulated upon rhizobacterial inoculation, with CEP peptides promoting nodulation [55,68,192], Recent data show that *Mt*CEP1 promotes AM symbiosis, while the reduced expression of *Sl*CEP2 during inoculation with AM fungi leads to an increase in lateral root formation [72,73]. This indicates that CEP peptides regulate symbiosis with both fungi and rhizobacteria.

Several peptide families exhibit overlapping regulatory functions in plant symbiotic interactions with both symbiotic bacteria and other rhizosphere organisms. This suggests a degree of conservation in the mechanisms underlying these relationships. Despite this, the distinct roles observed for some peptides in different symbiotic contexts highlight the complexity and specificity of these interactions. 

## 6. Conclusions and Future Perspectives

The recruitment of symbiotic partners offers significant benefits to the plant, including an enhanced availability of mineral nutrients, improved tolerance to stress conditions, and increased resistance to pathogens. In turn, the plant adapts its morphology to accommodate the symbiont, synthesizes specific metabolites, controls symbiont colonization and spread, and modulates its immune response accordingly. All these processes require coordinated and rapid regulation involving peptides. 

The contribution of different peptide groups to symbiosis regulation has been studied unevenly: cysteine-rich and PTM peptides are relatively well understood, whereas peptides encoded by smORFs and non-cysteine-rich, non-PTM peptides remain poorly explored, and no involvement of cryptic peptides has been identified. 

Most of the peptides that control symbiotic relationships appear to have had other functions initially. Primarily, this involves the regulation of root architecture and immunity. Peptide families that are involved in plant beneficial interactions can be divided into four categories according to their functions (Table 1). The CLE and CEP antagonistic families are involved in root growth regulation and nutrient uptake. NCR, SNARP/LP, and ENOD40 are exclusive to the development of indeterminate nodules during rhizobacterial symbiosis. Other groups, including RALF, PSK, nsLTP, GRP, RGF, DVL, and DEF, are considered to regulate plant interactions with soil organisms. Among them, RALF and PSK display a wide diversity of functions: they both control root growth, immune responses, and the establishment of beneficial relationships with rhizobacteria and PGPB. The more limited functional diversity observed in the other five families is likely due to insufficient data. Thus, further research may provide valuable insights into plant-beneficial organism interactions. Of particular interest is how the plant distinguishes between beneficial and pathogenic organisms, especially considering that differences may occur at the strain or species level, and how this may subsequently modulate its immune response. 

Peptide regulation has been most extensively studied in the context of the interaction between IRLC legumes and rhizobacteria, while other types of symbiotic relationships have received comparatively less attention. However, there is reason to believe that the same peptide families may participate in the regulation of various types of symbiotic interactions (Figure 3). Moreover, several plant pathogens produce mimetics of plant peptides to facilitate infection. Among beneficial organisms, peptide mimetics have so far only been found in the mycorrhizal fungi *Rhizophagus* and *Gigaspora* (Table 1).

Since symbionts supply the plant with deficient macronutrients and can also increase resistance to various stresses, understanding the mechanisms of cross-kingdom relationships is extremely valuable for the development of agrotechnologies. Components of symbiosis-controlling signaling modules, including peptides, are proposed to be used as targets for genetic engineering to create more productive plant varieties [193]. Peptides involved in the regulation of cross-kingdom relationships can be used for plant treatments or as an addition to microbial fertilizers [89]. Thus, the study of the role of regulatory peptides in the control of beneficial interactions is a very promising area from both fundamental and agricultural points of view.

## Figures and Tables

**Figure 1 plants-14-00378-f001:**
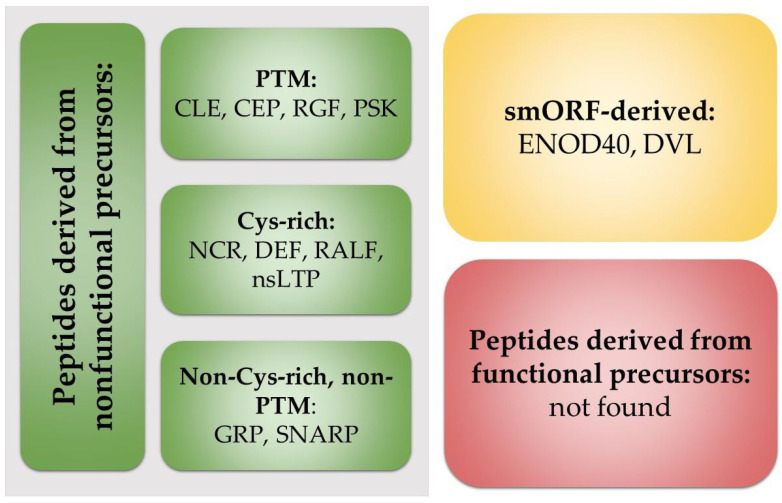
Classification of peptides regulating beneficial interactions.

**Figure 2 plants-14-00378-f002:**
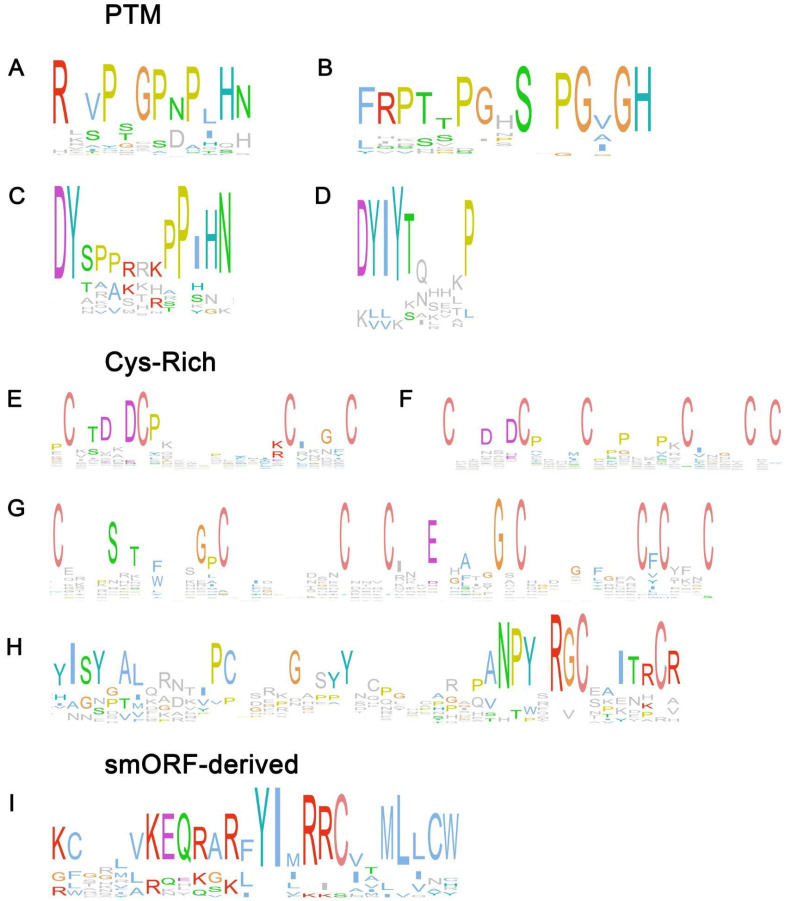
Conservative motifs of peptide families involved in the regulation of symbiosis in *Medicago truncatula*. (**A**) CLE, (**B**) CEP, (**C**) RGF, (**D**) PSK, (**E**) NCR-A, (**F**) NCR-B, (**G**) DEF, (**H**) RALF, and (**I**) DVL. To obtain the conservative motifs of plant SSP families, we used data about known SSPs for *Medicago truncatula* retrieved from MtSSPdb (https://mtsspdb.zhaolab.org/database/ accessed on 6 December 2024). The Mafft L-INS-I algorithm [41] was used for multiple sequence alignment, and the results were visualized using Jalview software 2.11.4.1 [42]. Clustalx coloring was applied.

**Figure 3 plants-14-00378-f003:**
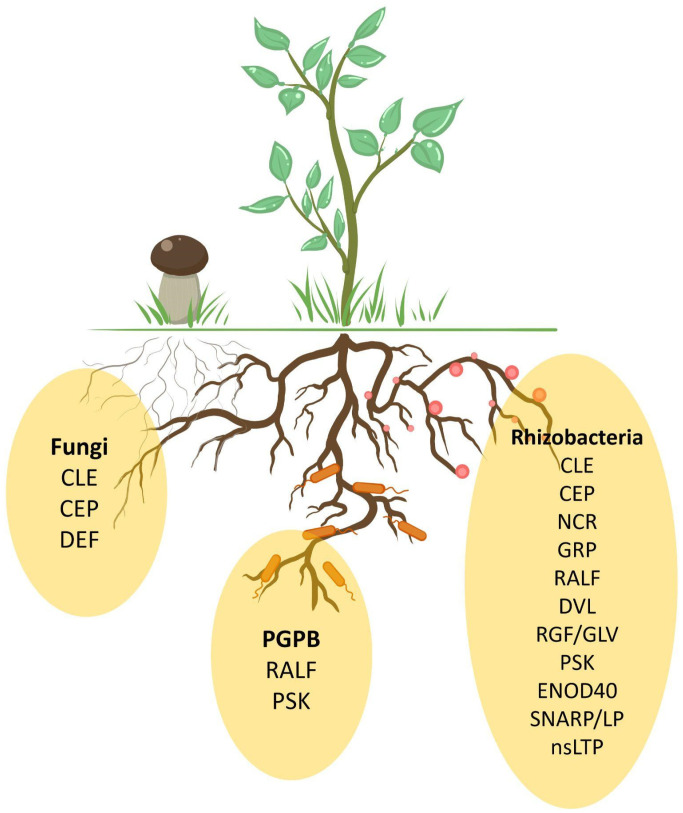
Peptide families regulating plant beneficial interactions with different rhizosphere organisms: rhizobacteria, fungi, and PGPBs.

**Table 1 plants-14-00378-t001:** Peptides involved in the regulation of beneficial interactions.

Peptide Family	Nodules	Mycorrhiza	PGPB	Produced by Pathogens	Root Growth	Immunity	References
CLE	+	+		+	+		[28,36,48,49]
CEP	+	+		+	+		[69,70,72,177]
RALF	+		+	+	+	+	[124,125,129,130,135,136,137]
PSK	+		+ (PSKR1)	+	+	+	[82,83,84,88,178]
nsLTP	+				+	+	[139,142,175]
GRP	+				+	+	[146,148,150]
RGF/GLV	+				+	+	[77,79,81]
DVL	+				+		[130,158]
DEF		+				+	[93,94]
NCR	+						[7,9,10]
SNARP/LP	+						[153]
ENOD40	+						[160]

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
