# Peer review of "The Small Key to the Treasure Chest: Endogenous Plant Peptides Involved in Symbiotic Interactions"

_plants, 2025, doi:10.3390/plants14030378_

Round 1

Reviewer 1 Report

Comments and Suggestions for Authors

Dear Authors,

I believe the authors have written a good article; this field is very broad, and they have made a great effort in synthesizing the information. However, the structure needs to be improved, especially section 2.3. The figure I suggested seems necessary to enhance understanding, and there are some important gaps in knowledge that are neither considered nor discussed.

Majors:

-L10 and L19, and L30, ……: “ bacteria and fungi” and microalgae too. Please note that microalgae are also found in roots interacting, and there is substantial literature on this subject.

-L96.....: Please explain in this section the distinguishing features that CLE peptides must have and what differentiates them from others.

-What is the homology between the different CLE peptides? Please discuss it.

-The different peptides explained in section 2 are not presented in order; that is, some from one family are explained, and then it jumps to another family. It would be better to explain all the peptides from one family first and then those from the other family, please.

-L197: "Mature NCRs consist of 35-55 a.a. and contain 4 or 6 conserved cysteine residues." Is the percentage of homology between these peptides known? What is the structure of the conserved cysteine domain like.

-L255-L263: This paragraph would fit better at the beginning of section 2.3, as it seems too general to be placed at the end.

-L265: "DEFs are immune CRPs, relative to NCRs." What type of similarity do they have?

-I believe it would greatly enhance the understanding of the paper if the authors prepared a figure that clearly highlights the structural differences between the various peptides in section 2.3, including regions of homology, size, and mode of action.

-L236: "PTMP" – Please ensure consistency; in some places, it is referred to as PTM, and in others as PTMP

-Please specify, when known, the mechanisms by which the plant secretes these peptides into the environment. Are there specific transporters? Is it through vehicles? Or is it unknown?

-Table 1 should be completed; a column indicating the references used is missing, please.

-L466: "Plant rhizosphere organisms have evolved mechanisms to manipulate host plants." Some microalga as Chlamydomonas has an L-amino acid oxidase on its membrane, which has been proposed to alter peptides and amino acids producing IAA. Please, discuss this.

-L495: “mimicry” sorry I don’t understand this term?

-In my opinion, the conclusions are too lengthy and therefore do not fulfill their purpose effectively. Please make them more concise

Reviewer 2 Report

Comments and Suggestions for Authors

In this manuscript the authors set out to provide a summary of the current literature regarding the roles specific families of plant peptides play in regulating plant-microbe symbiotic interactions. The literature review is on the whole thorough and up-to-date and the manuscript is generally written to an adequate standard. Overall, I found the manuscript to be broadly interesting, especially with regards to the lesser known peptides regulating symbiosis and the existence of trans-kingdom peptide mimicry by microbes to exploit host symbiotic control mechanisms. With a modest amount of changes, the manuscript would be suitable for publication in Plants although there is definitely further scope for improvement beyond this.

Major comments:

1) The majority of the literature assessed in this manuscript has also been covered semi-recently in an outstanding review article; Roy and Müller (2022) Trends in Plant Science. This manuscript assesses the same literature from very similar angles. The organization of literature, critical insight and quality of writing of the review by Roy and Müller is objectively superior to the current manuscript in its present state. This isn’t to say the present manuscript is not of publishable quality, only that it isn’t up to the incredibly high standard of the aforementioned review and that this may limit its value significantly. The need for a subsequent review can always be justified by the fact that new literature will have arisen since the review by Roy and Müller. However, the authors should be aware this raises the bar of quality for their work if they want it to be widely cited.

2) It is clear that the authors have engaged with a tremendous quantity of literature in writing this review. Whilst this has been generally done to a good standard, there is room for improvement here. First of all, there are a number of studies that I believe are either essential or at least highly relevant to scope of the review set out by the authors. For convenience, I have listed these in the essential changes of my report. Secondly, there are rare instances where the authors have misinterpreted the conclusions of some of the literature they are citing in their writing. These are also noted in the essential changes list although these mostly relate to the literature this reviewer is most familiar with and I cannot guarantee that this captures all instances of this. Finally, there are a number of occasions where literature is cited that is relevant to the scope of this review but in inappropriate contexts. These are detailed in other comments in my reviewer report.

Whilst the level of written English throughout the manuscript is generally reasonable, there are some parts where this needs to be improved. Whilst responsibility for detailed proof reading lies with the authors and not the reviewer, I appreciate that the authors may not be native English speakers and I feel that simply stating this needs improvement is neither constructive nor actionable feedback. Therefore at the end of this reviewer report, I include a separate list of suggested edits for the authors to improve the readability of their manuscript.

Although the scope of this review encompasses any plant peptide that has regulatory activity in the context of symbiotic interactions, the vast majority of the literature relates to nitrogen fixing symbiosis between legumes and rhizobia. There is also a significant amount of literature relating to interactions between AM fungi and their hosts. A much smaller quantity of literature is included relating to interactions between plants and their microbiomes. For this latter category, it is disputable if all of these interactions are truly symbiotic in nature. The definition of a symbiosis is a mutually beneficial close association between two or more organisms.

The vast majority of this review is highly descriptive. Consequentially, there are some areas that lack critical insight and the threads of all the literature in the manuscript are not always brought back into a cohesive narrative and outstanding questions are left unidentified. Whilst the authors do attempt this in sections 3-5, it is certainly not done to the same standard as in the review by Roy and Müller. One things that stands out to this reviewer is the enormous body of evidence for regulation of nodulation by small peptides in Medicago spp. relative to other legume species, especially those which form determinant nodules. Is this a consequence of lack of study in other models or does this offer an insight into how the most extreme levels of bacteroid differentiation that characterize indeterminant nodules evolved?

Essential changes: These are the minimum changes to the manuscript that are required to bring it up to a level suitable for publication in this journal and mostly relate to making the work as comprehensive and accurate as possible.

[Throughout manuscript] The authors stated aim of this review is to “provide a comprehensive summary of peptide families that facilitate beneficial relationships between plants and rhizosphere organisms”. Given this scope, if this review is to be truly comprehensive, then there is some literature absent that I would expect should be included. Whilst I appreciate there may be space constraints, all of the literature I list below is of outstanding relevance to the scope of this work:

NCR peptides;

Horvath et al., (2023) New Phytologist

Montiel et al., (2017) PNAS

Kim et al., (2015) PNAS

Zhang et al., (2023) Plant Physiology

Horvath et al., (2015) PNAS

Yang et al., (2017) PNAS

Wang et al., (2017) PNAS

Price et al., (2015) PNAS

CLE peptides;

Mens et al., (2021) New Phytologist

Mortier et al., (2010) Plant Physiology

Lebedeva et al, (2022) Agronomy

Lebedeva et al., (2023) International Journal of Molecular Sciences

Lebedeva et al., (2024) Journal of Plant Growth and Regulation

ENOD40;

Wan et al., (2007) Journal of Experimental Botany

Xu et al., (2020) New Phytologist

Li et al., (2022) PLOS Biology.

Please note that this is the literature related only to NCR, CLE peptides and ENOD40, where this reviewer’s expertise lies most strongly, and that there may be other literature that is omitted from this review.

[Throughout manuscript] Within this review and especially in the case of legume-rhizobia symbiosis, it is rather difficult to understand what is generally representative of these symbiotic interactions and what is specific to the context of a given plant-symbiont pairing. This can be addressed by rewriting sections to frame this – I would suggest in all major subsections of section 2, the authors employ a consistent format with an opening paragraph introducing the peptide family and its general role in plant regulation. This can then be followed with description of the available literature, one area at a time (e.g. in indeterminant then determinant nodules or discuss literature regarding the peptide then its receptor) and finish with a summary paragraph.

29-31: These are examples of symbiosis but not an accurate definition. The term symbiosis refers to a mutually beneficial and close association between two or more biological organisms – it is not specific to plant-microbe interactions.

138: This is surely not the correct reference and needs to be replaced with the correct one.

286: Does this statement refer to a specific subfamily of GRP peptides? From the review cited in this manuscript by Mangeon and colleagues, it should be apparent that GRPs are present in the genome of many non-legumes species, including Arabidopsis. Therefore the claim here that these are unique to IRLC legumes cannot be correct?

349-352: Whilst it is appreciated that the authors are simply reciting the conclusions of the study by Wang and colleagues here, how can it be that the increased nodule number seen in the transient PSK1/PSKR1 overexpression experiments could occur in the absence of increased nodule initiation when the latter is a perquisite of the former? At the time the nodule quantification was performed in those experiments, the absolute number of nodule primordia may not have shown significant statistical differences between the overexpressing lines and their respective empty vector controls. However, the statistically significant increase in mature/developing nodules would have been preceded by increased nodule initiation. The conclusion of the original study is inaccurate, but the authors of this manuscript do not need to propagate this inaccuracy.

381-384: This section should be rewritten to provide more clarity. At present, it does not accurately reflect the findings of the cited study by Ganguly and colleagues or our current knowledge of ENOD40 (see previous comment regarding missing literature). ENOD40 has been demonstrated as a positive regulator of nodule formation in multiple legume species including Medicago, soy bean and peanut. The activity of the antisense transcript of ENOD40 reported by Ganguly and colleagues is curious. Its relevance to a review on peptides is disputable however and to the best of this reviewer’s knowledge, this is unique to the peanut ENOD40 genes. Most importantly, it is not a conclusion of the aforementioned study that the RNA silencing of ASHR3 leads to a reduction in nodule formation. Possibly this stems from a misunderstanding of figure 1 of the Ganguly and colleagues study – this demonstrates a reduction in nodule numbers when ENOD40 is silenced using RNAi but this refers to RNA interference from an ad-hoc construct to knockdown expression of ENOD40 and not the by the naturally occurring ENOD40 antisense mRNA. Furthermore, the “80%/fivefold” reduction described in this study and propagated to the present manuscript is inaccurate – based off figure 1C, this is evidently a ~50% reduction.

528-533: The premise of this section seems to be the contrast between the activity of the tomato and Medicago CEPs during AM symbiosis, however this premise is flawed. The CEP2 peptide in tomato studied by Hsieh and colleagues is negatively regulated during AM symbiosis but there is not compelling evidence that it is itself a negative regulator of that symbiosis. For this reason, I suggest this section is either rewritten or removed from the review.

Further suggested edits: These are changes that I do not consider essential for the publication of the authors’ work in this journal but I do feel collectively would represent a significant improvement to their manuscript and its use/citability to the community of researchers of symbiotic plant-microbe interactions more on par with the review by Roy and Müller. Therefore, whilst I recommend incorporation of these changes I leave discretion regarding their implementation to the editor and authors.

35-38: Whilst invasion via root hairs is the most common entry mechanism for rhizobia into legume roots, it is not universal. Other mechanisms, such as crack entry, exist for rhizobia to infiltrate host roots. Indeed, some legume-rhizobia interactions have been reported that are independent of Nod factors. Therefore, for maximum accuracy, the authors may wish to amend this sentence; “Typically, flavonoid compounds secreted by the plant […]”.

41-42: From the flow of this section, it is not clear that this sentence is referring to all legume-diazotroph interactions rather than specifically Frankia.

41-42 (also 438-442): These references are too specific to give the reader an overview of the species specificity in N-fixing symbiosis. A more general reference would be more appropriate e.g. Walker et al., (2020) Frontiers in Microbiology.

45-49: Both of the references are also rather focused works that do not collectively capture all of the beneficial aspects of mycorrhizal symbiosis the authors discuss here. Once again, the authors should cite a more general reference that fully supports their statement e.g. Bennett and Groten, (2022) Annual Review of Plant Biology.

59-63: What is the relevance of the study by Zhang and colleagues to the authors’ statement here? The statement is referring to the beneficial effects of the microbiome whilst this study looks at the distribution of mycorrhizal fungi in a specific terrestrial habitat.

64-69: Both of these references are focused specifically on interactions between M. sativa and its symbionts but are supporting a statement that is construed to be generally representative of legume-diazotroph interactions. I would suggest that the authors begin by referring the reader to a more general overview of the transcriptomic response of legumes to their symbionts e.g. Mergaert et al., (2020), The Plant Cell prior to the more specific literature.

106-108: Some of these references are also very specific for what is intended to be an overview reference of autoregulation of nodulation.

146-148: Some reframing here would also be helpful – the first sentence states that CEPs are found in many seed plant species and is immediately followed by discussion of their roles in regulating nodule formation which clearly is not their role in the many non-legume seed plants.

190-193: This statement needs a reference.

197-198: There are more appropriate references than the one chosen here for an overview of NCR structure.

199-206: Whilst the literature cited and discussed here is undoubtedly very interesting, it is not clear how this work looking at the antimicrobial activity of synthetic (i.e. that do not occur in nature) peptides fits in the scope of this review. Therefore, the authors may want to consider its omission.

207-208: This statement lacks clarity and needs significantly rewording. My presumption is that the authors mean that the nodules of legume species whose genomes encode larger numbers of NCR family peptides tend to exhibit more extreme TBD phenotypes. But to call this a correlation is misleading – TBD is not really quantifiable – and G. uralensis (7) and M. truncatula (>700) possess significantly different numbers of NCR encoding genes – but the extent of bacteroid differentiation in the latter would not be 100 times the former.

208-211: This is excessively speculative. Lots of other genetic differences (besides NCR number) would exist between the two species and this study inoculates both species with the same strain of rhizobia – the opposite result may be found if the authors employed a different compatible strain.

219-222: Whilst one would expect a high degree of redundancy if such a large family of genes (>700 NCRs in M. truncatula), it should be noted that in the study referenced here, Güngör and colleagues only tested interactions with a single rhizobial strain in all but one of the NCR knockout lines. Given that there is a large body of evidence of single NCR genes being essential for symbiosis in some Medicago-rhizobia pairings and completely dispensable in others, the author’s conclusion that these NCR peptides have no effect on symbiosis is excessive.

230-232: These NCR-evasion mechanisms should not be framed simply as coping strategies which rhizobia can use to adapt to participate in normal symbiosis. In some cases, previously compatible legume-rhizobia pairings become incompatible (i.e. fix- phenotype) as a direct consequence of these mechanisms. In other cases, these mechanisms are employed by rhizobia with poor N-fixation ability (“cheaters”) to escape mechanisms of host selection and sequester host resources whilst providing little benefit in return.

236-237: Is there evidence for the widespread usage of peptidases to counteract the effect of host peptides outside of select interactions with NCRs in Medicago truncatula and its compatible symbionts? If not, then this statement may need to be revised.

297-301: This does not really provide any insight into the role of RALF peptides in regulating symbiosis for the reason the authors state; there is no experimental validation that these differentially expressed peptides have important effects (positive or negative) on symbiosis. Therefore the authors may want to consider removing this part.

326-342: This section cites the recent study by Roy and colleagues and discusses some of its findings. However, discussion of the role of GLV10 (described in this same study) in regulation the contextual positioning of nodules is not discussed, which would surely be in the scope of this manuscript.

386-391: This section would benefit from a bit of a rewrite to better frame the introduction of SNARPs/LPs. The existence of the family was first reported in Graham et al., 2004 Plant Physiology.

402: Is it the case that SNARPs are unique to species forming indeterminant nodules as stated here or unique to Medicago spp. as stated earlier? According to Trujillo and colleagues, there is no evidence for their existence outside of Medicago spp.

425-453: Though not strictly relevant to immunity or symbiosis, if the authors consider it relevant to their review they may wish to consider Greifenhagen et al., (2024) PNAS in this section as well.

436-437: The authors may consider mentioning effector-triggered immunity here – this is a layer of plant immunity that has evolved to directly target the presence of harmful effectors secreted by pathogens. In the case of commensal/symbiotic organisms, there are not selective forces that favour the recognition of effectors.

478-480: Given that bona fide symbionts bring benefits to their plant hosts, is it not evolutionarily intuitive that plants would evolve to recognize and associate with these without the need for peptide mimicry?

Typos/grammatical edits:

25: Suggest edit “The plants use beneficial […]” to “Plants use beneficial […]”

31-32: Suggest edit “The symbiotic microorganisms […]” to “These symbiotic microorganisms […].”

34: “Leguminosae” should not be accented (I think this could be a quirk of translation?).

48: Suggest edit “[…] pathogens, and insects […]” to “[…] pathogens and insects […]” (remove comma).

102: Suggest edit “[…] variable domain, and a C-terminal […]” to “[…] variable domain and a C-terminal […]” (remove comma).

124: There needs to be a space before reference 43 in this sentence.

146-147: Suggest editing of this sentence along the lines of “C-terminally encoded peptides (CEPs) are a multigene family present in seed plants.”

170: Suggest editing of this sentence or merging with the previous sentence as it does not make sense by itself like this.

171: This sentence is ambiguous. With its present wording it could be interpreted as suggesting that both CLE and CE family peptides are inhibitors of nodulation when the authors really mean that they are antagonists of each other. Therefore this sentence should be edited along the lines of “To conclude, CLE and CEP groups both regulate the extent of nodule formation through antagonistic mechanisms.”

173-176: These two sentences should be merged as they are essentially repetitive.

245-246: Suggest remove word “recently” from this sentence.

280-281: Suggest edit along the lines of “In Medicago truncatula and Medicago sativa, GRPs with nodule-specific expression patterns have been identified.”

286: In addition to containing two typos (“GPR” and “IRCL”), this sentence doesn’t read very well at all – suggest edit along the lines of “Similarly to the NCR family peptides, the GRP family peptides are present in legumes only amongst IRLC members.”

328-329: Suggest edit “[…] land plants that form roots or root-like structures” to “all land plants”. The authors seem to have misunderstood the first figure of the study by Roy and colleagues which states that GLV peptides are absent in the green algae but present in all root/rhizoid bearing plants (all major lineages of land plants have roots/rhizoids).

340: Suggest edit “Meanwhile, leave-expressed […]” to “Meanwhile, leaf-expressed […]”.

406-407: What is meant by “transferring activity”? Do the authors mean “transferase activity” instead?

411-413: Suggest rewrite along the lines of “In species capable of legume-rhizobia symbiosis, a significant proportion of nsLTPs in their genomes are induced during nodulation.”

425: Suggest edit “Simbiosis” to “Symbiosis”.

448: Suggest edit “sustainable” to “susceptible”. Additionally, there needs to be a space prior to the citation on this line.

467-468: Suggest edit “One of the mechanisms that decreased immune response is to […]” to “One mechanism these microbes can employ to dampen immune responses and facilitate these interactions is to […]”.

493: Suggest edit “In sum […]” to “In summary […]”.

541-542: Suggest edit: “[…] improved tolerance to stress conditions, and increased resistance to pathogens.” to “[…] improved tolerance to stress conditions and increased resistance to pathogens.” (remove comma)

563-564: Suggest edit “[…] differences may occur at the strain-specie level, and how it subsequently modulates its immune response.” to “[…] differences may occur at the strain or species level and how this may subsequently modulate its immune response.”

Reference formatting: Some citations in bibliography contain what looks like HTML e.g. [115]. Once the authors are certain that no further references need to be added, they should check through every reference to ensure the formatting is all correct.

Round 2

Reviewer 1 Report

Comments and Suggestions for Authors

I believe the authors have adequately addressed all of my comments and suggestions.

Reviewer 2 Report

Comments and Suggestions for Authors

The authors have satisfactorily addressed all my concerns from my initial comments, including the majority of ones that I considered ideal but not essential for publication. Consequentially, I feel the manuscript is significantly improved and I hope the authors would agree with this sentiment also.

Please note I noticed a handful of typos that have been introduced with the revisions that should be addressed ahead of final proofs.

44: Suggest edit "different zones of the nodule"

109: Suggest edit "Conservative motifs" to "Conserved motifs"

256: Suggest edit "pairing" to "pairings".